# Visualization of endogenous G proteins on endosomes and other organelles

**Wonjo Jang, Kanishka Senarath, Gavin Feinberg, Sumin Lu, Nevin A Lambert***

Department of Pharmacology and Toxicology, Medical College of Georgia, Augusta University, Augusta, United States

## eLife assessment

This **important** study investigates the intracellular localization patterns of G proteins involved in GPCR signaling, presenting **compelling** evidence for their preference for plasma and lysosomal membranes over endosomal, endoplasmic reticulum, and Golgi membranes. This discovery has significant implications for understanding GPCR action and signaling from intracellular locations. This research will interest cell biologists studying protein trafficking and pharmacologists exploring localized signaling phenomena.

*For correspondence:
NELAMBERT@augusta.edu

**Competing interest:** The authors declare that no competing interests exist.

**Abstract** Classical G-protein-coupled receptor (GPCR) signaling takes place in response to extracellular stimuli and involves receptors and heterotrimeric G proteins located at the plasma membrane. It has recently been established that GPCR signaling can also take place from intracellular membrane compartments, including endosomes that contain internalized receptors and ligands. While the mechanisms of GPCR endocytosis are well understood, it is not clear how well internalized receptors are supplied with G proteins. To address this gap, we use gene editing, confocal microscopy, and bioluminescence resonance energy transfer to study the distribution and trafficking of endogenous G proteins. We show here that constitutive endocytosis is sufficient to supply newly internalized endocytic vesicles with 20–30% of the G protein density found at the plasma membrane. We find that G proteins are present on early, late, and recycling endosomes, are abundant on lysosomes, but are virtually undetectable on the endoplasmic reticulum, mitochondria, and the medial-trans Golgi apparatus. Receptor activation does not change heterotrimer abundance on endosomes. Our findings provide a subcellular map of endogenous G protein distribution, suggest that G proteins may be partially excluded from nascent endocytic vesicles, and are likely to have implications for GPCR signaling from endosomes and other intracellular compartments.

## Introduction

Heterotrimeric G proteins transduce a vast number of important physiological signals (*Gilman, 1987*), most often in response to activation by G-protein-coupled receptors (GPCRs; *Pierce et al., 2002*). Canonical G protein signaling occurs when a cell surface GPCR is activated by an extracellular ligand, which in turn promotes activation of plasma membrane G protein heterotrimers and downstream effectors. Recently, it has become clear that GPCRs can also signal from intracellular compartments (*Calebiro et al., 2010*; *Eichel and von Zastrow, 2018*), most notably endosomes and the Golgi apparatus (*Calebiro et al., 2009*; *Ferrandon et al., 2009*; *Irannejad et al., 2017*; *Irannejad et al., 2013*; *Mullershausen et al., 2009*). Signaling from endosomes is often a continuation of signaling that starts at the plasma membrane and persists as (or resumes after) active receptors are endocytosed (*Tsvetanova et al., 2015*). Much is known about the machinery responsible for GPCR internalization, and also about the trafficking itineraries of specific receptors after endocytosis. Some receptors are efficiently

sorted for recycling and are returned to the plasma membrane, whereas other receptors are rapidly degraded (*Hanyaloglu and von Zastrow, 2008*). It is also known that at least one isoform of the G protein effector adenylyl cyclase is actively internalized (*Lazar et al., 2020*).

In contrast, much less is known about how G protein heterotrimers traffic from the plasma membrane through intracellular compartments (*Wedegaertner, 2012*). It is not known how efficiently heterotrimers are loaded onto endocytic vesicles at the plasma membrane, how receptor activation might change this process, or what the fate of G proteins might be after endocytosis. Activation at the plasma membrane promotes heterotrimer dissociation, and the resulting loss of membrane avidity allows Gβγ dimers and some Gα subunits (most notably Gα$_s$) to translocate through the cytosol to sample intracellular membranes (*Akgoz et al., 2004*; *Hynes et al., 2004*; *Ransnäs et al., 1989*; *Slepak and Hurley, 2008*; *Wedegaertner et al., 1996*). However, these processes reverse quickly when activation ceases (*Akgoz et al., 2004*; *Martin and Lambert, 2016*), meaning that activation-dependent translocation of free Gα subunits and Gβγ dimers would be an inefficient mechanism to deliver inactive heterotrimers to intracellular membranes. G proteins have been detected on the surface of endosomes and other intracellular compartments using a variety of approaches (*Hewav-itharana and Wedegaertner, 2012*; *Irannejad et al., 2013*; *Scarselli and Donaldson, 2009*; *Wede-gaertner, 2012*). However, there has been no quantitative comparison of G protein distribution across identified subcellular compartments. Studies of endogenous G proteins are limited by the availability of well-validated antibodies suitable for immunostaining, and overexpression of tagged G protein subunits may lead to aberrant localization.

Here, we study the subcellular distribution of endogenous heterotrimeric G proteins in cultured cells using CRISPR-mediated gene editing, confocal imaging, and bioluminescence resonance energy transfer (BRET). We find that G proteins are abundant on membrane compartments that are

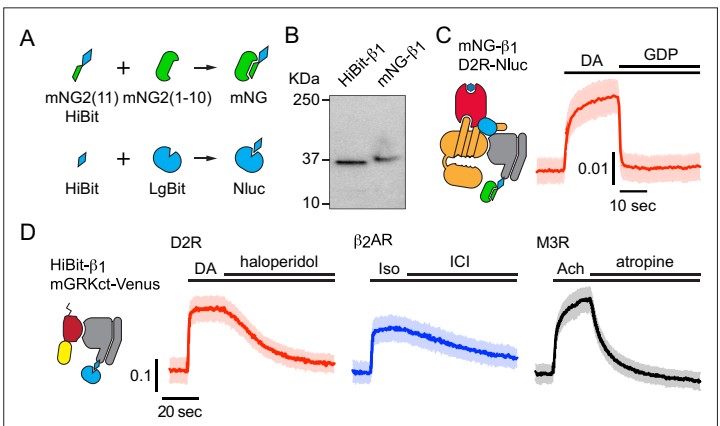

**Figure 1.** Validation of mNG-β$_1$ and HiBit-β$_1$ cells. (**A**) Cartoon showing the peptide tag complementation systems used to label endogenous Gβ$_1$ subunits. (**B**) SDS-PAGE of HiBit-β$_1$ and mNG-β$_1$ cell lysates; the predicted molecular weights of the edited gene products are 38.9 and 41.1 kilodaltons (KDa), respectively; representative of 3 independent experiments. (**C**) In permeabilized nucleotide-depleted cells BRET between dopamine D2R-Nluc receptors and mNG-β$_1$-containing heterotrimers increases in response to dopamine (DA; 100 μM) and reverses after addition of GDP (100 μM); mean ± 95% CI; n=27 replicates from two independent experiments. (**D**) In intact cells BRET between HiBit-β$_1$ and the Gβγ sensor memGRKct-Venus increases after stimulation of D2R dopamine, β$_2$AR adrenergic, or M3R acetylcholine receptors with DA (100 μM), isoproterenol (Iso; 10 μM) and acetylcholine (Ach; 100 μM), respectively. Signals reversed when receptors were blocked with haloperidol (10 μM), ICI 118551 (10 μM) or atropine (10 μM); mean ± 95% CI; n=16 replicates from four independent experiments.

The online version of this article includes the following source data and figure supplement(s) for figure 1:

**Source data 1.** PDF file containing original HiBit blot shown in panel B, indicating the relevant bands.

**Source data 2.** Original files for HiBit blot shown in panel B.

**Source data 3.** Numerical data for traces shown in panels C and D.

**Figure supplement 1.** Receptor-mediated accumulation of cyclic AMP (cAMP) is similar in HiBit-β$_1$, mNG-β$_1$ and parent cell lines.

**Figure supplement 1—source data 1.** Numerical data for traces shown in panels A and B.

functionally continuous with the plasma membrane, including early, late, and recycling endosomes. However, heterotrimer density on endocytic membranes is lower than on the plasma membrane, suggesting that G protein endocytosis is inefficient. Endocytic trafficking of G proteins is not regulated by GPCRs. Our findings are likely to have implications for GPCR signaling from endosomes, as internalized receptors are concentrated in G-protein-deficient compartments.

## Results

To study the localization of endogenous G proteins, we used gene editing to attach small peptide tags to the amino terminus of $G\beta_1$ subunits (*GNB1*) in HEK 293 cells. We chose this subunit because it is the most abundant $G\beta$ subunit in this cell type (*Cho et al., 2022*), it can associate with any type of $G\alpha$ or $G\gamma$ subunit (*Hillenbrand et al., 2015*), and it can be labeled at this position without disrupting heterotrimer formation or function. For bioluminescence experiments we added the HiBit tag (*Schwinn et al., 2018*) and isolated clonal 'HiBit-$\beta_1$' cell lines. An advantage of this approach over adding a full-length Nanoluc luciferase is that it requires coexpression of LgBit to produce a complemented luciferase. This limits luminescence to cotransfected cells and thus eliminates background from untransfected cells. For imaging experiments we added a tandem tag that included the 11th beta strand of mNeonGreen2 (mNG2(11); *Feng et al., 2017*) and HiBit in cells constitutively expressing mNG2(1–10) and isolated 'mNG-$\beta_1$' cell lines (*Figure 1A*). Amplicon sequencing verified that cell lines had correctly edited *GNB1* genes and SDS-PAGE revealed single proteins with apparent molecular weights consistent with edited $G\beta_1$ subunits (*Figure 1B*). BRET assays demonstrated that tagged subunits in HiBit-$\beta_1$ and mNG-$\beta_1$ cell lines formed functional heterotrimers with endogenous $G\alpha$ and $G\gamma$ subunits. For example, we observed agonist-induced BRET between the D2 dopamine receptor and mNG-$\beta_1$, an interaction that requires association with endogenous $G\alpha$ subunits (*Figure 1C*). Similarly, we observed BRET between HiBit-$\beta_1$ and the free $G\beta\gamma$ sensor memGRKct-Venus after activation of receptors that couple $G_{i/o}$, $G_s$, and $G_q$ heterotrimers, indicating that HiBit-$\beta_1$ associated with endogenous $G\alpha$ subunits from these three families (*Figure 1D*). We also found that cyclic AMP accumulation in response to stimulation of endogenous β adrenergic receptors was similar in edited cell lines and their unedited parent lines (*Figure 1—figure supplement 1*). Endogenous $G\alpha$ and $G\beta$ subunits are expressed at approximately a 1:1 ratio, and $G\beta$ subunits are tightly associated with $G\gamma$ and inactive $G\alpha$ subunits (*Cho et al., 2022*; *Gilman, 1987*; *Krumins and Gilman, 2006*). Moreover, proteins that bind to free $G\beta\gamma$ dimers are found in the cytosol of unstimulated HEK 293 cells, suggesting at most only a small population of free $G\beta\gamma$ in these cells (*Barak et al., 1999*). Therefore, we assume that almost all mNG-$\beta_1$ and HiBit-$\beta_1$ subunits in unstimulated cells are part of heterotrimers.

### Endogenous G proteins primarily associate with the plasma membrane and endolysosomes

Confocal imaging of mNG-$\beta_1$ cells revealed the expected bright fluorescence at the plasma membrane. Most cells also contained pleiomorphic intracellular structures and dim cytosolic fluorescence that was sufficient to suggest relative exclusion of mNG-$\beta_1$ from the nucleus. Especially notable were clusters of large vesicular structures located at the cell periphery which were later identified as lysosomes (*Figure 2A*; see below). Large intracellular organelles such as the endoplasmic reticulum, mitochondria, and Golgi apparatus were not evident. Unsurprisingly, our images are quite similar to those made as part of previous study that labeled $G\beta_1$ subunits with mNG2 (*Cho et al., 2022*).

To identify the intracellular membrane compartments with mNG-$\beta_1$ fluorescence, we coexpressed a series of organelle markers tagged with red fluorescent proteins. Markers of the endoplasmic reticulum, mitochondria and medial-trans Golgi apparatus indicated that these large compartments were virtually devoid of mNG-$\beta_1$ fluorescence (*Figure 2B–D*, *Figure 2—figure supplements 1–3*). In some cells, an indistinct region of mNG-$\beta_1$ fluorescence was interleaved with leaflets of the Golgi apparatus, but line profiles suggested that this was a distinct structure (*Figure 2D*, *Figure 2—figure supplement 3*), most likely the perinuclear recycling compartment (see below).

In contrast, mNG-$\beta_1$ clearly colocalized with the marker FYVE, which binds to phosphatidylinositol-3-phosphate (PI3P) on the surface of endosomes (*Figure 3A and B*, *Figure 3—figure supplement 1*). However, mNG-$\beta_1$ fluorescence was not detected on every FYVE-positive vesicle and when present was much less intense than fluorescence of adjacent segments of the plasma membrane (*Figure 3B*).

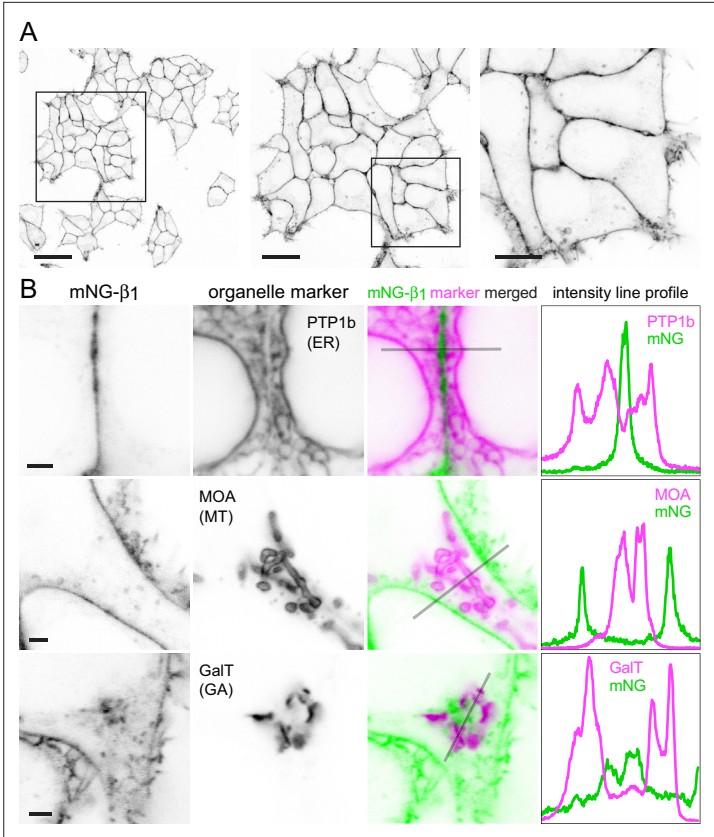

**Figure 2.** Endogenous G proteins are abundant on the plasma membrane but not large organelles. (**A**) A single field of view of mNG-$\beta_1$ cells at three magnifications; scale bars are 40 μm, 20 μm, and 10 μm. (**B**) mNG-$\beta_1$ does not colocalize with expressed markers of the endoplasmic reticulum (ER; PTP1b), mitochondria (MT; MOA) or medial-trans Golgi apparatus (GA; GalT); intensity line profiles depict absolute fluorescence intensity in each channel; scale bars are 2 μm.

The online version of this article includes the following source data and figure supplement(s) for figure 2:

**Figure supplement 1.** G proteins are not abundant on the endoplasmic reticulum (ER).

**Figure supplement 2.** G proteins are not abundant on mitochondria (MT).

**Figure supplement 2—source data 1.** Numerical data for individual line scans (panel C).

**Figure supplement 3.** G proteins are not abundant on the medial-trans Golgi apparatus (GA).

The median signal-to-background ratio for FYVE-positive structures was less than one-fourth that of the plasma membrane (*Figure 3C*). FYVE domains primarily localize to early endosomes (*Hammond and Balla, 2015*), so it was not surprising that similar colocalization of mNG-$\beta_1$ was observed with the early endosome marker rab5a. As was the case with FYVE, mNG-$\beta_1$ was detectable in some rab5a-positive vesicles but not others and was not as intense as the nearby plasma membrane (*Figure 3A and C*, *Figure 3—figure supplement 2*). In order to determine the fate of G proteins after endocytosis, we then examined mNG-$\beta_1$ colocalization with markers of recycling and late endosomes (*Stenmark, 2009*). Dim mNG-$\beta_1$ fluorescence was detected on indistinct rab11a-positive structures clustered diffusely in the vicinity of the nucleus (*Figure 3A*, *Figure 3—figure supplement 3*), which we presumptively identified as the perinuclear recycling compartment (PNRC). Similarly, mNG-$\beta_1$ colocalized extensively with vesicles labeled with rab7a, a marker of late endosomes (*Figure 3A*, *Figure 3—figure supplement 4*). Notably, mNG-$\beta_1$ fluorescence was more intense on rab7a-positive late endosomes than on FYVE- or rab5a-positive early endosomes (*Figure 3C*). The presence of mNG-$\beta_1$ on late endosomes suggested that some G proteins may be degraded by lysosomes. Accordingly, mNG-$\beta_1$ strongly colocalized with lysosomes marked with LysoView 633 (*Figure 3A*, *Figure 3—figure supplement 5*), long-term incubation with fluorescent dextran (*Figure 3—figure supplement 5*), or the lysosome marker LAMP1 (*Figure 3—figure supplement 6*). While some mNG fluorescence was detected in the lumen

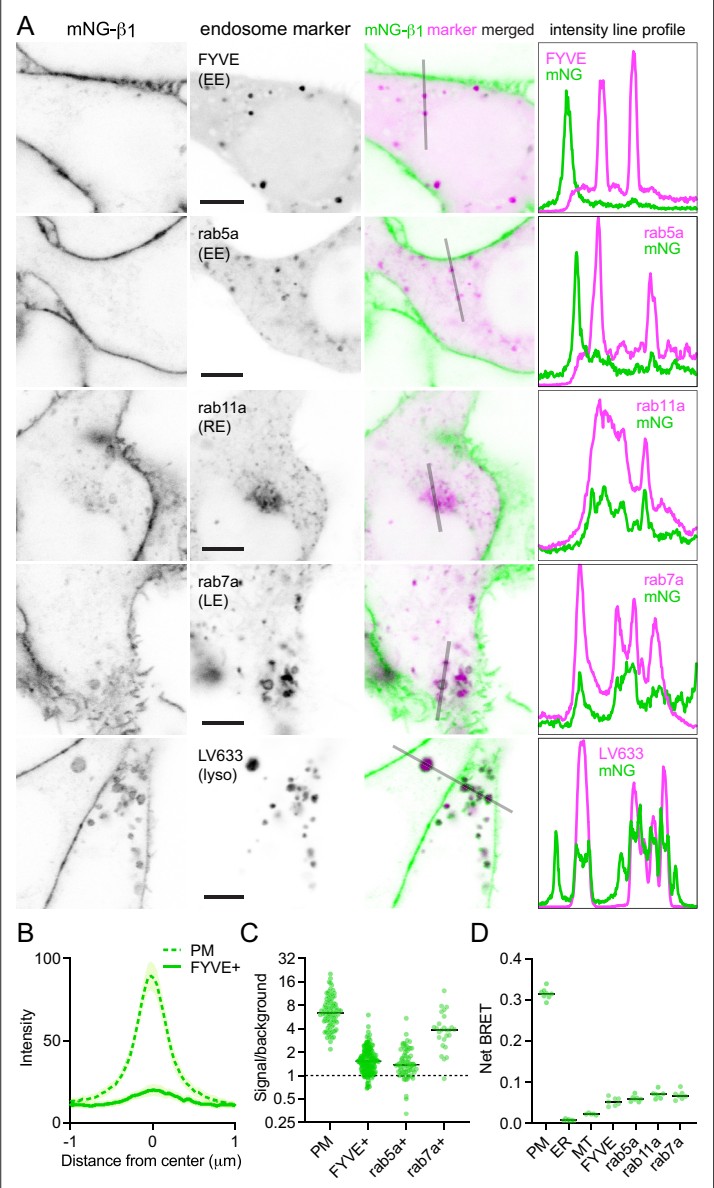

**Figure 3.** Endogenous G proteins colocalize with markers of endosomes and lysosomes. (**A**) mNG-$\beta_1$ colocalizes with expressed markers of early endosomes (EE; FYVE and rab5a), recycling endosomes (RE; rab11a), late endosomes (LE; rab7a) and lysosomes (lyso; LysoView 633); intensity line profiles depict absolute fluorescence intensity in each channel; scale bars are 5 µm. (**B**) Mean mNG-$\beta_1$ fluorescence intensity line profiles drawn across the plasma membrane (PM) and FYVE-positive vesicles; mean ±95% CI; n=40 vesicles/cells. (**C**) mNG-$\beta_1$ signal/background ratios for regions of interest surrounding the plasma membrane (PM; n=99), FYVE-positive (n=125) and rab5a-positive (n=56) early endosomes, and rab7a-positive (n=26) late endosomes; horizontal lines represent the median. (**D**) Bystander net BRET signals between HiBit-$\beta_1$ and Venus-tagged markers of the plasma membrane (PM), endoplasmic reticulum (ER), mitochondria (MT), early endosomes (FYVE and rab5a), recycling endosomes (rab11a) and late endosomes (rab7a); horizontal lines represent the median; n=5–7 independent experiments.

The online version of this article includes the following source data and figure supplement(s) for figure 3:

**Source data 1.** Numerical data for line profiles (panel B), signal/background ratios (panel C) and bystander BRET (panel D).

**Figure supplement 1.** G proteins colocalize with the early endosome marker FYVE on some endosomes.

**Figure supplement 2.** G proteins colocalize with the early endosome marker rab5a on some endosomes.

**Figure supplement 3.** G proteins colocalize with the recycling endosome marker rab11a in a perinuclear region.

*Figure 3 continued*

**Figure supplement 4.** G proteins colocalize with the late endosome marker rab7a on many endosomes.

**Figure supplement 5.** G proteins are abundant on lysosomes.

**Figure supplement 6.** G proteins colocalize with the lysosome marker LAMP1.

**Figure supplement 7.** G proteins colocalize with the trans-Golgi marker TGNP.

of lysosomes, much of the signal remained on the cytosolic surface of these structures, and in many instances the intensity of mNG-$\beta_1$ fluorescence on lysosomes was similar to that of the nearby plasma membrane (*Figure 3A*). We also detected robust mNG-$\beta_1$ signals on structures labeled with TGNP (*Figure 3—figure supplement 7*), a marker of the trans-Golgi network. These imaging results suggest that G proteins are likely to undergo endocytosis and enter both recycling and degradative pathways and may become more concentrated as late endosomes mature.

As an alternative approach, we performed bystander BRET experiments to map the subcellular localization of endogenous HiBit-$\beta_1$. This approach provides an unbiased index of membrane protein colocalization from large populations of cells and has the additional advantage of very high sensitivity (*Lan et al., 2012*). We expressed LgBit and a series of inert Venus-tagged membrane markers in HiBit-$\beta_1$ cells and observed large bystander signals at the plasma membrane, smaller bystander signals at endosomes, and very small bystander signals at the endoplasmic reticulum and mitochondria (*Figure 3D*). Although bystander BRET signals cannot be directly compared between different compartments, these results are generally consistent with what we observed using confocal imaging and confirm the presence of G proteins on multiple endosomal compartments.

## Constitutive G protein endocytosis is inefficient

That mNG-$\beta_1$ fluorescence was less intense on endosomes than the plasma membrane suggested that G protein density may be lower on the surface of endosomes than on the plasma membrane.

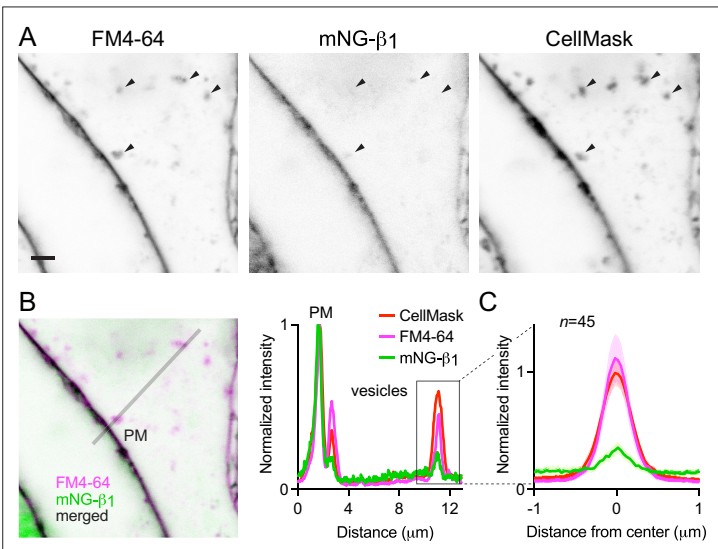

**Figure 4.** Constitutive G protein endocytosis is inefficient. (**A**) mNG-$\beta_1$ colocalizes with newly internalized endocytic vesicles labeled with FM4-64 and CellMask Deep Red (arrowheads); scale bar is 2 µm. (**B**) A fluorescence intensity line profile for mNG-$\beta_1$, FM4-64 and CellMask normalized to the peak value of each label at the plasma membrane (PM). (**C**) Mean mNG-$\beta_1$, FM4-64 and CellMask fluorescence intensity line profiles drawn across vesicles, normalized to fluorescence intensity at the plasma membrane for each label; mean ± 95% CI; n=45 vesicles/cells.

The online version of this article includes the following source data and figure supplement(s) for figure 4:

**Source data 1.** Numerical data for line profiles (panel C).

**Figure supplement 1.** Constitutive endocytosis of G proteins is inefficient.

**Figure supplement 2.** mNG-HRas ct is less abundant on endocytic vesicles than the plasma membrane.

**Figure supplement 2—source data 1.** Numerical data for line profiles (panel C).

However, differences in fluorescence intensity could also be due to differences in the amount of membrane surface area sampled in the imaging volume. Likewise, differences in bystander BRET between compartments could be due to differences in several factors, including expression and efficiency of compartment-specific BRET acceptors. Therefore, we devised a co-labeling protocol that allowed us to compare mNG-$\beta_1$ fluorescence to the amount of newly internalized membrane imaged at endocytic vesicles, and to simultaneously make the same measurements at the plasma membrane. To stain both the plasma membrane as well as newly formed endocytic vesicle membrane, we exposed live cells to the styryl dye FM4-64, which rapidly and reversibly partitions into (but does not cross) membranes and is only fluorescent in a hydrophobic environment (*Betz et al., 1996*). When cells are exposed to FM4-64 at physiological temperatures the plasma membrane is stained immediately, and this is followed over the course of several minutes by the appearance of intracellular vesicles that have trapped the dye (*Figure 4A*, *Figure 4—figure supplement 1*). As an orthogonal approach we stained cells with CellMask Deep Red, a lipophilic dye that permanently stains the plasma membrane and therefore is incorporated into endocytic vesicles. Both dyes are expected to produce fluorescence signals proportional to the surface area of the membrane sampled by the imaging method, allowing us to normalize the fluorescence of individual vesicles to the nearby plasma membrane. We reasoned that if G proteins are passively incorporated into endocytic vesicles without any enrichment or exclusion, then mNG-$\beta_1$ fluorescence in each vesicle should have the same intensity relative to the plasma membrane as lipophilic dyes. After staining cells and allowing 15 min for constitutive endocytosis, we found that FM4-64 and CellMask dyes reported similar amounts of membrane surface area in endocytic vesicles (*Figure 4A and B*); in both cases, peak vesicle intensity was on average similar to the intensity of the plasma membrane (*Figure 4C*). In contrast, peak mNG-$\beta_1$ fluorescence on the same endocytic vesicles was much less intense than the plasma membrane (*Figure 4A–C*). There was considerable variability between individual vesicles, such that some vesicles contained no detectable mNG-$\beta_1$ fluorescence (*Figure 4A*, *Figure 4—figure supplement 1*). Using FM4-64 fluorescence as a standard for membrane surface area, we calculated that mNG-$\beta_1$ density on FM4-64-positive vesicles was 20 ± 8% (mean ± 95% CI; n=45) of the nearby plasma membrane. This result confirms that heterotrimeric G proteins are present on newly internalized membrane but also suggests that G proteins are partially excluded from endocytic vesicles. To test if other peripheral membrane proteins are similarly depleted from endocytic vesicles, we performed analogous experiments by overexpressing mNG bearing the C-terminal membrane anchor of HRas (mNG-HRas ct). We found that mNG-HRas ct was also less abundant on FM4-64-positive endocytic vesicles than on the plasma membrane, although not to the same extent as mNG-$\beta_1$ (*Figure 4—figure supplement 2*); mNG-HRas ct density on FM4-64-positive vesicles was 64 ± 17% (mean ± 95% CI; n=78) of the nearby plasma membrane.

## Receptor activation does not change G protein endocytosis

The above results suggested that constitutive endocytosis of heterotrimeric G proteins is inefficient. However, it is possible that GPCR and G protein activation could change how G proteins are loaded onto endocytic vesicles. To test this possibility, we performed similar imaging experiments with mNG-$\beta_1$ cells transfected with SNAP-tagged $\beta_2$ adrenergic receptors (SNAPf-$\beta_2$AR). This receptor is often used as a model of activity-dependent GPCR internalization (*Benovic et al., 1988*; *von Zastrow and Kobilka, 1992*) and has been shown to activate G proteins on endosomes (*Bowman et al., 2016*; *Irannejad et al., 2013*). We labeled SNAPf-$\beta_2$AR with a membrane-impermeant SNAP ligand (AF 647) at room temperature to prevent constitutive endocytosis, then incubated cells with FM4-64 and the agonist isoproterenol for 15 min at physiological temperature to promote receptor endocytosis. Confocal imaging after agonist washout revealed numerous intracellular vesicles with intense AF 647 fluorescence, consistent with robust receptor internalization (*Figure 5A*). Normalization and comparison to FM4-64 fluorescence indicated that SNAPf-$\beta_2$AR was enriched approximately threefold on endocytic vesicles compared to the nearby plasma membrane (*Figure 5B and C*), consistent with active recruitment of active receptors to clathrin-coated pits and endocytic vesicles. In contrast, mNG-$\beta_1$ fluorescence in the same vesicles was again lower than expected given the amount of membrane imaged in each vesicle (*Figure 5B and C*). Once again there was considerable variability between individual endocytic vesicles (*Figure 5A and D*). Using FM4-64 fluorescence as a standard for membrane surface area, we calculated that mNG-$\beta_1$ density on receptor-containing vesicles was 28 ± 8% (mean ±95% CI; n=91) of the nearby plasma membrane. Although this density was higher

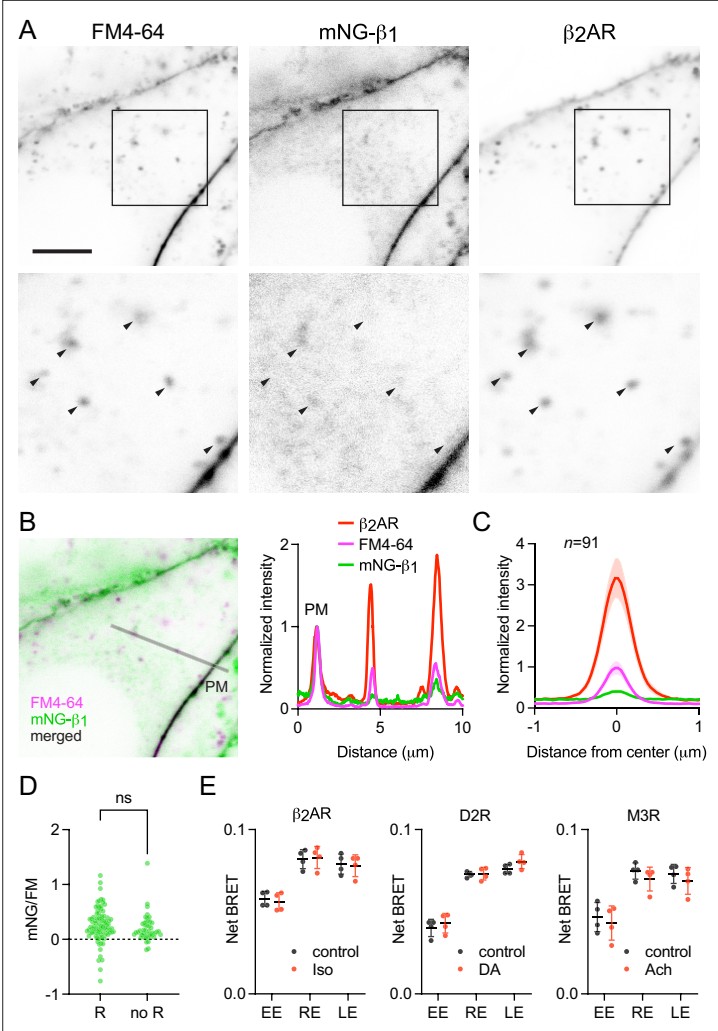

**Figure 5.** Receptor activation does not change G protein endocytosis. (**A**) mNG-$\beta_1$ colocalizes with newly internalized endocytic vesicles labeled with FM4-64 and SNAP-tagged $\beta_2$ adrenergic receptor ($\beta_2$AR) labeled with Alexa Fluor 674; scale bar is 5 µm. Cells were stimulated with 10 µM isoproterenol for 15 min to induce $\beta_2$AR internalization. (**B**) A fluorescence intensity line profile for mNG-$\beta_1$, FM4-64 and $\beta_2$AR normalized to the peak value of each label at the plasma membrane (PM). (**C**) Mean mNG-$\beta_1$, FM4-64 and $\beta_2$AR fluorescence intensity line profiles drawn across vesicles, normalized to fluorescence intensity at the plasma membrane for each marker; mean ± 95% CI; n=91 vesicles/cells. (**D**) Normalized peak mNG-$\beta_1$/FM4-64 did not differ between vesicles that contained receptors (R; n=91) and vesicles formed by constitutive endocytosis (no R; n=45); n.s., not significant, p=0.20, unpaired t-test. (**E**) Bystander BRET between HiBit-$\beta_1$ and markers of early endosomes (EE), recycling endosomes (RE) and late endosomes (LE) was unchanged after 30 minutes of receptor activation with isoproterenol (Iso; 10 µM), dopamine (DA; 100 µM) or acetylcholine (Ach; 100 µM) compared to the DPBS vehicle alone (control); mean ± SD, n=4 independent experiments; no agonist-treated group was significantly different from the control, paired t-test with a false discovery rate (FDR) of 1%.

The online version of this article includes the following source data and figure supplement(s) for figure 5:

**Source data 1.** Numerical data for line profiles (panel C), intensity ratios (panel D) and bystander BRET (panel E).

**Figure supplement 1.** Transient translocation of endogenous HiBit-$\beta_1$ from the plasma membrane to intracellular compartments *during* activation.

**Figure supplement 1—source data 1.** Numerical data for bystander BRET values (panel A) and traces (panel B).

**Figure supplement 2.** G protein abundance on endosomes after GPCR and G protein activation.

**Figure supplement 2—source data 1.** Numerical data for bystander BRET (panels A and B).

than that calculated for vesicles formed by constitutive endocytosis, the difference did not reach significance (*Figure 5D*). These results demonstrate that activation-dependent internalization of $\beta_2$ adrenergic receptors does not significantly promote or prevent loading of G proteins onto endocytic vesicles.

These findings suggested that $\beta_2$ adrenergic receptor activation should have no impact on the abundance of heterotrimeric G proteins on endosomes. To test this idea for this and other receptors, we performed bystander BRET experiments with HiBit-$\beta_1$ cells transiently expressing $\beta_2$ adrenergic, D2 dopamine or M3 muscarinic receptors to activate $G_s$, $G_{i/o}$, and $G_{q/11}$ heterotrimers, respectively. With overexpressed G proteins free G$\beta\gamma$ dimers translocate from the plasma membrane and sample intracellular membrane compartments when dissociated from G$\alpha$ subunits. Consistent with this, we observed small decreases in bystander BRET at the plasma membrane and small increases in bystander BRET at intracellular compartments during activation of GPCRs, suggesting that endogenous G$\beta\gamma$ subunits undergo similar translocation (*Figure 5—figure supplement 1*). Notably, these changes occurred at room temperature, suggesting that endocytosis was not involved, and developed over the course of minutes. The latter observation and the small magnitude of agonist-induced changes are both consistent with expression of primarily slowly-translocating endogenous G$\gamma$ subtypes in HEK 293 cells. Moreover, as shown previously for overexpressed G$\beta\gamma$, the changes we observed with endogenous G$\beta\gamma$ were readily reversible (*Figure 5—figure supplement 1*), suggesting that most heterotrimers reassemble at the plasma membrane after activation ceases. In order to assess changes in heterotrimer trafficking to endosomes without interference from G$\beta\gamma$ translocation we incubated cells with agonist under conditions permissive for vesicular trafficking, then washed with antagonist to inactivate receptors and allow heterotrimers to reassemble prior to measuring BRET. Under these conditions no significant changes in bystander BRET were observed at any endosome compartment after 30 min of receptor stimulation (*Figure 5E*). Similar results were obtained when stimulation was maintained for 15 min or 5 min (*Figure 5—figure supplement 2*), although small increases were apparent in a few instances. Taken together these results indicate that activation of receptors and G proteins does not substantially change heterotrimer abundance on the surface of endosomes.

## Discussion

While the mechanisms involved in the biosynthesis, chaperoning and trafficking of nascent G protein heterotrimers are fairly well understood (*Gabay et al., 2011*; *Marrari et al., 2007*; *Wedegaertner, 2012*), the mechanisms that regulate the subcellular distribution of heterotrimers after delivery to the plasma membrane have not been studied as extensively. Here, we show that constitutive and activity-dependent endocytosis of G proteins is inefficient. Avoidance of endocytosis is likely to be important for maintaining a high density of heterotrimers at the plasma membrane, where much important signaling takes place. On the other hand, this limits the abundance of G proteins on the surface of endosomes. At present we can only speculate regarding the mechanism that limits G protein density on endocytic vesicles. Many endocytosis mechanisms, including clathrin-mediated endocytosis, rely on bulky coat proteins and adapters to induce membrane curvature and recruit cargo (*Doherty and McMahon, 2009*). One possibility is that heterotrimeric G proteins are simply excluded from nascent endocytic vesicles by steric occlusion. While large extracellular domains are known to impede endocytosis of membrane proteins (*DeGroot et al., 2018*), a similar relationship has not been demonstrated for intracellular domains. Our finding that the small mNG-HRas ct probe is loaded onto endocytic vesicles more efficiently than mNG-$\beta_1$-containing heterotrimers is consistent with this idea. It is noteworthy that the monomeric G proteins HRas and NRas are also less abundant on endosomes than the plasma membrane, and therefore are separated from internalized growth factor receptors (*Pinilla-Macua et al., 2016*; *Surve et al., 2021*).

Some studies using overexpressed G protein subunits have suggested that a large pool of G proteins is located on intracellular membranes, including the Golgi apparatus (*Chisari et al., 2007*; *Saini et al., 2007*; *Tsutsumi et al., 2009*), whereas others have indicated a distribution that is dominated by the plasma membrane (*Crouthamel et al., 2008*; *Evanko et al., 2000*; *Marrari et al., 2007*; *Takida and Wedegaertner, 2003*). A likely factor contributing to these discrepant results is the stoichiometry of overexpressed subunits, as neither G$\alpha$ nor G$\beta\gamma$ traffic appropriately to the plasma membrane as free subunits (*Wedegaertner, 2012*). Our gene-editing approach presumably maintains the native subunit stoichiometry, providing a more accurate representation of native G protein distribution. Our results

show that endogenous G proteins are primarily located on the plasma membrane and are present on internal membranes at substantially lower levels. We identify the specific intracellular compartments where G proteins are found and show the relative abundance of G proteins on each compartment. Nascent heterotrimers are likely formed and lipid modified on the endoplasmic reticulum and Golgi apparatus (*Wedegaertner, 2012*), yet few heterotrimers can be found on these compartments at any given moment, consistent with a relatively slow rate of turnover compared to forward trafficking during biosynthesis (*Gabay et al., 2011*). Notably, when Gβγ dimers are expressed alone they accumulate on the endoplasmic reticulum (*Michaelson et al., 2002*; *Takida and Wedegaertner, 2003*). That we detect almost no endogenous Gβγ on the endoplasmic reticulum supports our conclusion that the large majority of Gβγ in unstimulated HEK 293 cells is associated with Gα. Likewise, we found that few heterotrimers are associated with mitochondria, despite the fact that previous studies have demonstrated functional roles for G proteins on these organelles (*Hewavitharana and Wedegaertner, 2012*). Our results suggest that GPCR signaling from intracellular compartments will generally have to be transduced by a lower density of G proteins. Consistent with these findings, a recent large-scale study assessing protein abundance on organelles in HEK 293 cells found that $Gβ_1$ is enriched on the plasma membrane and lysosomes but is not significantly enriched on endosomes, the Golgi apparatus, endoplasmic reticulum, or mitochondria (*Hein et al., 2023*).

Fully lipid-modified heterotrimers in their inactive state are unlikely to detach from membranes at a significant rate (*Shahinian and Silvius, 1995*). Therefore, we infer from the presence of $Gβ_1$ on early, late, and recycling endosomes that heterotrimers undergo vesicle-mediated endocytosis in unstimulated cells and are not efficiently sorted to either the slow recycling pathway or the degradative pathway. Our imaging results also show that G proteins are apparently more abundant on late endosomes and lysosomes than on early endosomes, suggesting that they become concentrated as late endosomes mature. We do not know the fate of G proteins located on the surface of lysosomes. Since lysosomes may fuse with the plasma membrane under certain circumstances (*Xu and Ren, 2015*), it is possible that this represents a route of G protein recycling to the plasma membrane. Our results are largely consistent with the hypothesis that G proteins passively follow bulk endocytic flow of membrane and suggest that at least some G proteins are recycled to the plasma membrane. We also cannot exclude the possibility that heterotrimers traffic between membrane compartments by mechanisms other than vesicular trafficking (*Saini et al., 2009*). However, even if this is the case our conclusions that G proteins are internalized inefficiently and are present at lower density on most intracellular membranes are still valid.

Tagging endogenous proteins with HiBit and other luciferase fragments has proven to be useful for studying several aspects of GPCR signaling (*White et al., 2020*). The cell lines we developed should prove useful for answering additional questions related to G protein regulation, such as possible non-vesicular translocation due to palmitate turnover (*Saini et al., 2009*; *Wedegaertner and Bourne, 1994*), the role of ubiquitination (*Dohlman and Campbell, 2019*), and localization in subcompartments not studied here. In addition, it may also be possible to use these cells in combination with targeted sensors to study endogenous G protein activation in different subcellular compartments. More broadly, our results show that subcellular localization of endogenous membrane proteins can be studied in living cells by adding a HiBit tag and performing bystander BRET mapping. Applied at large scale this approach would have some advantages over fluorescent protein complementation (*Cho et al., 2022*), most notably the ability to localize endogenous membrane proteins that are expressed at levels that are too low to permit fluorescence microscopy.

Our study is not without limitations. Our labeling strategy was designed to interfere as little as possible with heterotrimer function, but we cannot rule out the possibility that the tags we used to visualize and track G proteins had some influence on their trafficking. By labeling $Gβ_1$ subunits, we cannot directly distinguish heterotrimers from free Gβγ dimers, complicating interpretation. This strategy also does not allow us to resolve heterotrimers containing different Gα subunits. It is quite possible that heterotrimers containing different Gα subunits could be subject to different trafficking mechanisms. Our conclusion that GPCR activation has no lasting effect on the subcellular distribution of G proteins rests on three representative receptors, chosen because they activate three of the four major G protein families. It is possible that other receptors will influence G protein distribution using mechanisms not shared by the receptors we studied. For example, a few receptors are thought to form relatively stable complexes with Gβγ, which could provide a mechanism of trafficking to

endosomes (*Thomsen et al., 2016*; *Wehbi et al., 2013*). Finally, our study was limited to a single non-differentiated cell type. It would not be surprising to find that differentiated cells have mechanisms to regulate G protein trafficking and distribution that are not shared by the model cells we used (*Calebiro et al., 2009*; *Kotowski et al., 2011*; *Lin et al., 2024*; *Puri et al., 2022*).

In summary, here we show that heterotrimeric G proteins are more abundant on the plasma membrane than on any intracellular compartment where they are thought to be important for signaling. Our results are likely to have functional implications for signaling from intracellular compartments. Receptor-G protein coupling is thought to be rate-limited by collision and G protein abundance (*Hein et al., 2005*), and decreasing G protein expression is known to impair downstream signaling (*Gabay et al., 2011*; *Schwindinger et al., 1997*). Signaling from endosomes and other compartments may thus be disadvantaged by a low density of G proteins. Further studies are warranted to examine the stoichiometry of receptors, G proteins, regulators, and effectors in different subcellular compartments and how this affects signaling.

# Materials and methods

## Key resources table

| Reagent type (species) or resource | Designation | Source or reference | Identifiers | Additional information |
|---|---|---|---|---|
| Gene (*Homo sapiens*) | *GNB1* | GenBank | Gene ID: 2782 | Gene (*Homo sapiens*) |
| Cell line (*Homo sapiens*) | HEK293 | ATCC | CRL-1573; RRID:CVCL_0045 | Cell line (*Homo sapiens*) |
| Cell line (*Homo sapiens*) | HEK293T expressing mNG2(1–10) | PMID:35271311 | | Obtained from Manuel Leonetti |
| Cell line (*Homo sapiens*) | HiBit-β1 | This paper | | See Materials and Methods |
| Cell line (*Homo sapiens*) | mNG-β1 | This paper | | See Materials and Methods |
| Sequence-based reagent | crRNA | Integrated DNA Technologies | | See Materials and Methods |
| Sequence-based reagent | tracrRNA | Integrated DNA Technologies | | See Materials and Methods |
| Sequence-based reagent | ssODN HDR donor | Integrated DNA Technologies | | See Materials and Methods |
| Recombinant protein | Cas9 Nuclease V3 | Integrated DNA Technologies | 1081059 | |
| Recombinant DNA reagent | mRuby-Golgi-7 | Addgene | 55865 | |
| Recombinant DNA reagent | mRuby2-Rab5a-7 | Addgene | 55911 | |
| Recombinant DNA reagent | mCherry-Rab7a-7 | Addgene | 55127 | |
| Recombinant DNA reagent | mCherry-Rab11a-7 | Addgene | 55124 | |
| Recombinant DNA reagent | pmCherry-2xFYVE | Addgene | 140050 | |
| Recombinant DNA reagent | mCherry-TGNP-N-10 | Addgene | 55145 | |
| Recombinant DNA reagent | Lamp1-mScarlet-I | Addgene | 98827 | |
| Recombinant DNA reagent | Venus-2xFYVE | This paper | | Venus version of pmCherry-2xFYVE |
| Recombinant DNA reagent | mRuby2-MOA | This paper | | mRuby2 version of Venus-MOA |
| Recombinant DNA reagent | mRuby2-PTP1b | This paper | | mRuby2 version of Venus-PTP1b |
| Recombinant DNA reagent | Venus-kras | PMID:21364942 | | |
| Recombinant DNA reagent | Venus-PTP1b | PMID:22816793 | | |
| Recombinant DNA reagent | Venus-MOA | PMID:22816793 | | |
| Recombinant DNA reagent | Venus-rab5a | PMID:21364942 | | |
| Recombinant DNA reagent | Venus-rab7a | PMID:27528603 | | |
| Recombinant DNA reagent | Venus-rab11a | PMID:27528603 | | |
| Recombinant DNA reagent | memGRKct-Venus | PMID:19258039 | | |
| Chemical compound | PEI Max | Polysciences | 24765 | |
| Other | CellMask Deep Red | ThermoFisher | C10046 | 1:1,000 |
| Other | LysoView 633 | Biotium | 70058 | 1:1,000 |
| Other | CF 640 dextran | Biotium | 80115 | 25 µg ml$^{-1}$ |

*Continued on next page*

| Reagent type (species) or resource | Designation | Source or reference | Identifiers | Additional information |
| --- | --- | --- | --- | --- |
| Other | FM4-64; SynaptoRed | Sigma-Aldrich | 574799 | 5 µM |
| Software | CRISPResso2 | PMID:30809026 | RRID:SCR_024503 | |
| Software | ImageJ | imagej.net/ij/ | RRID:SCR_003070 | |
| Software | GraphPad Prism | graphpad.com | RRID:SCR_002798 | |

## Cell culture and transfection

Human embryonic kidney HEK 293 cells (ATCC; CRL-1573) were propagated in 100 mm dishes, on six-well plates, or on 25 mm round coverslips in high glucose DMEM (Cytiva) and 10% fetal bovine serum (Cytiva) supplemented with penicillin streptomycin (Gibco). HEK 293T cells stably expressing mNG2(1–10) (*Cho et al., 2022*) were kindly supplied by Manuel Leonetti (Chan Zuckerberg Biohub San Francisco). Cells were transfected in growth medium using linear polyethyleneimine MAX (Polysciences) at a nitrogen/phosphate ratio of 20 and were used for experiments 24–48 hr later. Up to 3.0 µg of plasmid DNA was transfected in each well of a six-well plate.

## Gene editing

Ribonucleoprotein (RNP) complexes were assembled in vitro in IDT nuclease free duplex buffer from Alt-R crRNA, Alt-R tracrRNA and Alt-R S.p. Cas9 Nuclease V3 purchased from Integrated DNA Technologies (IDT). RNP and repair ssODNs (dissolved in nuclease-free water) were added to single-cell suspensions (10 µl of $1.2 \times 10^4$ cells $µl^{-1}$) and electroporated using a Neon Transfection device (Invitrogen) following the manufacturer's instructions. Cells were expanded and diluted into 48-well plates and grown for 3 weeks. Wells containing single cell colonies were duplicated into 12-well plates and screened for HiBit expression by mixing crude lysates with purified LgBit protein (Promega) and measuring luminescence in the presence of 5 µM furimazine. After clone expansion genomic DNA was extracted using the GeneJET Genomic DNA purification kit (Thermo Fisher) and used as a template for amplicon sequencing. Sequencing primers were designed to span the editing site and to produce amplicons less than 500 base pairs in length. Amplicon sequencing was performed by Azenta Life Sciences (Amplicon-EZ) and analyzed using CRISPResso2 (*Clement et al., 2019*). Cell lines used for experiments had correctly edited alleles but were hemizygous due to competing repair mechanisms. The human *GNB1* gene was targeted at a site corresponding to the N-terminus of the G$\beta_1$ protein; the sequence 5'-TGAGTGAGCTTGACCAGTTA-3' was incorporated into the crRNA, and the same gRNA was used to produce both HiBit-$\beta_1$ and mNG-$\beta_1$ cell lines. The ssODN homology-directed repair (HDR) template sequence for mNG-$\beta_1$ cells was: ATCTCACATTCT TGAAGGTGGCATTGAAGAGCACTAAGATCGGAAGATG**ACCGAGCTCAACTTCAAGGAGTGG CAAAAGGCCTTTACCGATATGATG***GGCGGAAGCGGT***GTGTCCGGCTGGCGGCTGTTCAAG AAGATTTCT***GGCGGAAGC*AGTGAGCTTGACCAGCCTTAGACAGGAGGCCGAGCAACTTAAGAAC CAGA, with the mNG2(11) and HiBit tag sequences in **bold** font, and GGSG and GGS linkers in *italic* font. The repair template sequence for HiBit-$\beta_1$ cells was: TTTCAGATCTCACATTCTTGAAGG TGGCATTGAAGAGCACTAAGATCGGAAG**ATGGTGAGCGGCTGGCGGCTGTTCAAGAAGAT TAGC***GGCGGAAGCGGT*AGTGAGCTTGACCAGCTTAGACAGGAGGCCGAGCAACTTAAGAAC CAGATTCGAG, with the HiBit tag sequence in **bold** font, and GGSG linker in *italic* font. For both ssODNs a silent mutation (underlined sequence) was introduced to ablate the PAM site. HiBit was included in the repair template for producing mNG-$\beta_1$ cells to enable screening for edited clones using luminescence.

## SDS-PAGE

Pelleted cells were mixed with Laemmli buffer (Bio-Rad), and proteins were separated on 4 to 15% SDS polyacrylamide gradient gels (Bio-Rad) then transferred to polyvinylidene difluoride (PVDF) membranes (Millipore Sigma). HiBit-tagged proteins were detected using the NanoGlo HiBit Blotting kit (Promega) following the manufacturer's instructions, and membranes were imaged using an Amersham Imager 600.

## Plasmids

The following plasmids were used as received from Addgene: mRuby-Golgi-7 (GalT; #55865), mRuby2-Rab5a-7 (#55911), mCherry-Rab7a-7 (#55127), mCherry-Rab11a-7 (#55124), pmCherry-2xFYVE (#140050), mCherry-TGNP-N-10 (#55145), Lamp1-mScarlet-I (#98827). Venus-2xFYVE was made by replacing mCherry in pmCherry-2xFYVE with Venus using *NheI* and *BsrGI*. mRuby2-MOA was made by replacing Venus in Venus-MOA using *NheI* and *BglII*. mRuby2-PTP1b was made by replacing Venus in Venus-PTP1b using *NheI* and *BsrGI*. CMV-LgBit was made by amplifying LgBit from pBiT1.1-N (Promega) and ligating into pcDNA3.1 (+) using *HindIII* and *XhoI*. SNAPf-$\beta_2$AR, SNAPf-D2R, SNAPf-M3R and D2S-Nluc were kindly provided by Jonathan Javitch (Columbia University). The Nluc-EPAC-VV cyclic AMP sensor was kindly provided by Kirill Martemyanov (University of Florida). Venus-Kras, Venus-PTP1b, Venus-MOA, Venus-rab5a, Venus-rab7a, Venus-rab11a and memGRKct-Venus were described previously (*Hollins et al., 2009*; *Lan et al., 2012*). All plasmids were verified by automated sequencing.

## Imaging

Imaging was performed on a Leica SP8 laser scanning confocal microscope using a 63×1.40 NA oil immersion objective. Cells grown on 25 mm round coverslips were transferred to a steel imaging chamber and imaged in HEPES Imaging (HI) buffer which contained 150 mM NaCl, 10 mM NaHEPES, 5 mM glucose, 2.5 mM KCl, 1.2 mM $CaCl_2$, 1 mM $MgCl_2$ (pH 7.2). All imaging was carried out at room temperature with the exception of the experiment shown in *Figure 4—figure supplement 1*, which was carried out at 37 °C. For colocalization of mNG-$\beta_1$ and red organelle markers 0.2 μg of each marker was transfected per coverslip; mNG-β1 was excited at 488 nm and detected at 495–545 nm, and red markers were excited at 552 nm and detected at 565–665 nm. Lysosomes were stained with either LysoView 633 (Biotium; 1:1000 in growth medium for 15 min at 37 °C) or 10,000 m.w. CF 640 dextran (Biotium; 25 μg ml$^{-1}$ overnight at 37 °C followed by a 60 min chase); both dyes were excited at 638 nm and detected at 650–700 nm. For simultaneous imaging of mNG-$\beta_1$, FM4-64 and Cell-Mask Deep Red, cells were placed in HI buffer containing 1:1000 CellMask Deep Red (Invitrogen) for 15 min at room temperature, then returned to culture medium containing 5 μM FM4-64 (a.k.a. SynaptoRed; Calbiochem) and incubated at 37 °C for 15 min. Imaging was then performed in HI buffer containing 5 μM FM4-64. For simultaneous imaging of mNG-$\beta_1$, FM4-64 and $\beta_2$AR, cells were transfected with 1 μg of SNAPf-$\beta_2$AR, stained in HI buffer containing 5 μM SNAP-Surface Alexa 647 (New England Biolabs) for 15 min at room temperature, then returned to culture medium containing 10 μM isoproterenol and 5 μM FM4-64 and incubated at 37 °C for 15 min. Imaging was then performed in HI buffer containing 5 μM FM4-64. CellMask Deep Red and SNAP-Surface Alexa 647 were excited at 638 nm and detected at 650–700 nm; mNG-$\beta_1$ and FM4-64 were excited at 488 nm and detected at 500–570 nm and 675–755 nm, respectively.

## Image analysis

Signal/background ratios for the plasma membrane and endosomes (*Figure 3C*) were calculated using mean fluorescence values from rectangular (for the plasma membrane) and round (for endosomes) regions of interest (ROIs) surrounding the structures and nearby cytosol (for background). A single plasma membrane ROI and 1–3 endosome ROIs were sampled per cell/image. Mean fluorescence intensity line profiles were extracted from 3 μm lines centered on vesicles as absolute fluorescence intensity (*Figure 3B*), or fluorescence intensity normalized to the mean intensity of a nearby section of plasma membrane (*Figure 4C* and *Figure 5C*). Vesicles that contained and did not contain internalized receptors were compared (*Figure 5D*) by dividing the peak mNG-$\beta_1$ signal by the peak FM4-64 signal for each vesicle; both signals were first normalized to their respective plasma membrane signals and subjected to background subtraction. A single vesicle was sampled per cell/image. All image analysis was carried out using ImageJ and raw images. For construction of figures images were exported as.TIF files with or without uniform contrast enhancement applied by ImageJ.

## BRET

For bystander BRET mapping of HiBit-$\beta_1$ localization cells were transfected in 6-well plates with 1 μg per well of a Venus-tagged compartment marker and 0.1 μg per well of CMV-LgBit. For measurements cells were resuspended in Dulbecco's phosphate buffered saline (DPBS). For long-term agonist

stimulation (*Figure 5E*) HiBit-$\beta_1$ cells expressing CMV-LgBit (0.1 µg per well) and either SNAPf-$\beta$2AR, SNAPf-D2R or SNAPf-M3R (0.5 µg per well) were incubated with agonist for 30 min in the incubator, then washed and resuspended in DPBS containing antagonist prior to reading BRET. Agonists were isoproterenol (10 µM), dopamine (100 µM), and acetylcholine (100 µM); antagonists were ICI 118551 (10 µM), haloperidol (10 µM) and atropine (10 µM); all small molecule ligands were obtained from Millipore Sigma or Cayman Chemical. For functional validation of mNG-$\beta_1$ cells, D2R-Nluc (50 ng per well) was transfected, and cells were resuspended in permeabilization buffer (KPS) containing 140 mM KCl, 10 mM NaCl, 1 mM MgCl$_2$, 0.1 mM Potassium EGTA, 20 mM NaHEPES (pH 7.2), 10 µg ml$^{-1}$ high-purity digitonin and 2 U ml$^{-1}$ apyrase. Kinetic BRET measurements were made from permeabilized cells during sequential injection of dopamine (100 µM) and GDP (100 µM). For functional validation of HiBit-$\beta_1$ cells, SNAPf-$\beta$2AR, SNAPf-D2R or SNAPf-M3R (0.5 µg per well), CMV-LgBit (0.1 µg per well) and memGRK3ct-Venus (0.5 µg per well) were transfected, and cells were resuspended in DPBS. Kinetic BRET measurements were made from intact cells during sequential injection of agonists and antagonists at the concentrations listed above. All BRET measurements were made in buffer solutions containing the substrate furimazine (Promega or ChemShuttle; 1:1000 from a 5 mM stock dissolved in 90% ethanol/10% glycerol). Steady-state BRET and luminescence measurements were made using a Mithras LB940 photon-counting plate reader (Berthold Technologies GmbH) running MicroWin2000 software. Kinetic BRET measurements were made using a Polarstar Optima plate reader (BMG Labtech) running BMG Optima version 2.20R2 software. Raw BRET signals were calculated as the emission intensity at 520–545 nm divided by the emission intensity at 475–495 nm. Net BRET signals were calculated as the raw BRET signal minus the raw BRET signal measured from cells expressing only the donor.

## Statistical analysis

All statistical testing was carried out using GraphPad Prism version 10.1.1. Comparison of mNG signals in vesicles with and without receptors (*Figure 5D*) was made using an unpaired t-test. Comparison of endosome bystander signals with and without agonist treatment (*Figure 5E*) was made using paired t-tests with a false discovery rate (FDR) of 1% (method of Benjamini, Krieger and Yekutieli). Experiments are defined as independent biological replicates when performed on a separate passage of cells independently transfected or treated on different days.

## Materials availability

Cell lines and plasmids generated for this study are freely available without restriction upon request from the corresponding author. No unique code or software was used for the study.

## Acknowledgements

We thank Manuel Leonetti for providing HEK 293 cells expressing mNG2(1–10) and Kirill Martemyanov and Jonathan Javitch for providing plasmid DNA.

## Additional information

### Funding

| Funder | Grant reference number | Author |
| --- | --- | --- |
| National Institute of General Medical Sciences | GM145284 | Nevin A Lambert |

The funders had no role in study design, data collection and interpretation, or the decision to submit the work for publication.

### Author contributions

Wonjo Jang, Conceptualization, Investigation, Writing - review and editing; Kanishka Senarath, Gavin Feinberg, Formal analysis, Investigation; Sumin Lu, Resources, Investigation; Nevin A Lambert, Conceptualization, Funding acquisition, Investigation, Writing - original draft, Project administration

## Author ORCIDs
Wonjo Jang (iD) https://orcid.org/0000-0002-1928-8978
Gavin Feinberg (iD) http://orcid.org/0009-0002-7065-0917
Nevin A Lambert (iD) https://orcid.org/0000-0001-7550-0921

Reviewer #1 (Public review): https://doi.org/10.7554/eLife.97033.3.sa1
Reviewer #2 (Public review): https://doi.org/10.7554/eLife.97033.3.sa2
Reviewer #3 (Public review): https://doi.org/10.7554/eLife.97033.3.sa3
Author response https://doi.org/10.7554/eLife.97033.3.sa4

## Additional files

### Supplementary files
• MDAR checklist

### Data availability
All data generated or analyzed during this study are included in the manuscript and supporting files; source data files have been provided for *Figures 1, 3–5*, *Figure 1—figure supplement 1*, *Figure 4—figure supplement 2*, and *Figure 5—figure supplements 1 and 2*.

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
