## [Editor Report · eLife assessment]

This **important** study investigates the intracellular localization patterns of G proteins involved in GPCR signaling, presenting **compelling** evidence for their preference for plasma and lysosomal membranes over endosomal, endoplasmic reticulum, and Golgi membranes. This discovery has significant implications for understanding GPCR action and signaling from intracellular locations. This research will interest cell biologists studying protein trafficking and pharmacologists exploring localized signaling phenomena.

---

## [Referee Report · Reviewer #1 (Public review)]

Summary:

The manuscript by Jang et al. describes the application of new methods to measure the localization GTP-binding signaling proteins (G proteins) on different membrane structures in a model mammalian cell line (HEK293). G proteins mediate signaling by receptors found at the cell surface (GPCRs), with evidence from the last 15 years suggesting that GPCRs can induce G-protein mediated signaling from different membrane structures within the cell, with variation in signal localization leading to different cellular outcomes. While it has been clearly shown that different GPCRs efficiently traffic to various intracellular compartments, it is less clear whether G proteins traffic in the same manor, and whether GPCR trafficking facilitates "passenger" G protein trafficking. This question was a blind spot in the burgeoning field of GPCR localized signaling in need of careful study, and the results obtained will serve as an important guide post for further work in this field.

The extent to which G proteins localize to different membranes within the cell is the main experimental question tested in this manuscript. This question is pursued by through two distinct methods, both relying on genetic modification of the G-beta subunit with a tag. In one method, G-beta is modified with a small fragment of the fluorescent protein mNG, which combines with the larger mNG fragment to form a fully functional fluorescent protein to facilitate protein trafficking by fluorescent microscopy. This approach was combined with expression of fluorescent proteins directed to various intracellular compartments (different types of endosomes, lysosome, endoplasmic reticulum, golgi, mitochondria) to look for colocalization of G-beta with these markers. These experiments showed compelling evidence that G-beta co-localizes with markers at the plasma membrane and the lysosome, with weak or absent co-localization for other markers. A second method for measuring localization relied on fusing G-beta with a small fragment from a miniature luciferase (HiBit) that combines with a larger luciferase fragment (LgBit) to form an active luciferase enzyme. Localization of G-beta (and luciferase signal) was measured using a method known as bystander BRET, which relies on expression of a fluorescent protein BRET acceptor in different cellular compartments. Results using bystander BRET supported findings from fluorescence microscopy experiments. These methods for tracking G protein localization were also used to probe other questions. The activation of GPCRs from different classes had virtually no impact on the localization of G-beta, suggesting that GPCR activation does not result in shuttling of G proteins through the endosomal pathway with activated receptors.

In the revised version of this manuscript the authors have performed informative and important new experiments in addition to adding new text to address conceptual questions. These new data and discussions are commendable and address most or all of the weaknesses listed in the initial review.

Strengths:

The question probed in this study is quite important and, in my opinion, understudied by the pharmacology community. The results presented here are an important call to be cognizant of the localization of GPCR coupling partners in different cellular compartments. Abundant reports of endosomal GPCR signaling need to consider how the impact of lower G protein abundance on endosomal membranes will affect the signaling responses under study.

*The work presented is carefully executed, with seemingly high levels of technical rigor. These studies benefit from probing the experimental questions at hand using two different methods of measurement (fluorescent microscopy and bystander BRET). The observation that both methods arrive at the same (or a very similar) answer inspires confidence about the validity of these findings.

Weaknesses:

*As noted by the authors, they do not demonstrate that the tagged G-beta is predominantly found within heterotrimeric G protein complexes. In the revised manuscript the authors have added new discussion text on why it is likely that G-beta is mostly found in complexes. This line of reasoning is convincing, although more robust experimental methods for assessing the assembly status of G-beta could be a valuable target for future experimental developments.

---

## [Referee Report · Reviewer #2 (Public review)]

This study assess the subcellular distribution of a major G protein subunit (Gβ1) when expressed at an endogenous level in a well-studied model cell system (293 cells). The approach elegantly extends a gene editing strategy described by Leonetti's group and combines it with a FRET-based proximity assay to detect the presence of endogenously tagged Gβ1 on membrane compartments of 293 cells. The authors achieve their goal, and the data are convincing and interesting. The authors do a nice job of integrating their results with previous work in the field. The methods are now sufficiently well-described to enable other investigators to apply or adapt them in future studies.

---

## [Referee Report · Reviewer #3 (Public review)]

Summary:

This article addresses an important and interesting question concerning intracellular localization and dynamics of endogenous G proteins. The fate and trafficking of G protein-coupled receptors (GPCRs) have been extensively studied but so far little is known about the trafficking routes of their partner G proteins that are known to dissociate from their respective receptors upon activation of the signaling pathway. Authors utilize modern cell biology tools including genome editing and bystander bioluminescence resonance energy transfer (BRET) to probe intracellular localization of G proteins in various membrane compartments in steady state and also upon receptor activation. Data presented in this manuscript shows that while G proteins are mostly present on the plasma membrane, they can be also detected in endosomal compartments, especially in late endosomes and lysosomes. This distribution, according to data presented in this study, seems not to be affected by receptor activation. These findings will have implications in further studies addressing GPCR signaling mechanisms from intracellular compartments.

Strengths:

The methods used in this study are adequate for the question asked. Especially use of genome-edited cells (for addition of the tag on one of the G proteins) is a great choice to prevent effects of overexpression. Moreover, use of bystander BRET allowed authors to probe intracellular localization of G proteins in a very high-throughput fashion. By combining imaging and BRET authors convincingly show that G proteins are very low abundant on early endosomes (also ER, mitochondria, and medial Golgi), however seem to accumulate on membranes of late endosomal compartments. Moreover, authors also looked at the dynamics of G protein trafficking by tracking them over multiple time points in different compartments.

Weaknesses:

While authors provide a novel dataset, many questions regarding G protein trafficking remain open. For example, it is not entirely clear which pathway is utilized to traffic G proteins from the plasma membrane to intracellular compartments. Additionally, future studies should also include more quantitative details considering G-protein distribution in different compartments as well as more detailed dynamic data on G protein internalization as well as intracellular trafficking kinetics.

---

## [Author Response]

The following is the authors’ response to the original reviews.

**Public Reviews:**

**Reviewer #1 (Public Review):**
Summary:The manuscript by Jang et al. describes the application of new methods to measure the localization of GTP-binding signaling proteins (G proteins) on different membrane structures in a model mammalian cell line (HEK293). G proteins mediate signaling by receptors found at the cell surface (GPCRs), with evidence from the last 15 years suggesting that GPCRs can induce G-protein mediated signaling from different membrane structures within the cell, with variation in signal localization leading to different cellular outcomes. While it has been clearly shown that different GPCRs efficiently traffic to various intracellular compartments, it is less clear whether G proteins traffic in the same manner, and whether GPCR trafficking facilitates "passenger" G protein trafficking. This question was a blind spot in the burgeoning field of GPCR localized signaling in need of careful study, and the results obtained will serve as an important guidepost for further work in this field. The extent to which G proteins localize to different membranes within the cell is the main experimental question tested in this manuscript. This question is pursued through two distinct methods, both relying on genetic modification of the G-beta subunit with a tag. In one method, G-beta is modified with a small fragment of the fluorescent protein mNG, which combines with the larger mNG fragment to form a fully functional fluorescent protein to facilitate protein trafficking by fluorescent microscopy. This approach was combined with the expression of fluorescent proteins directed to various intracellular compartments (different types of endosomes, lysosome, endoplasmic reticulum, Golgi, mitochondria) to look for colocalization of G-beta with these markers. These experiments showed compelling evidence that G-beta co-localizes with markers at the plasma membrane and the lysosome, with weak or absent co-localization for other markers. A second method for measuring localization relied on fusing G-beta with a small fragment from a miniature luciferase (HiBit) that combines with a larger luciferase fragment (LgBit) to form an active luciferase enzyme. Localization of Gbeta (and luciferase signal) was measured using a method known as bystander BRET, which relies on the expression of a fluorescent protein BRET acceptor in different cellular compartments. Results using bystander BRET supported findings from fluorescence microscopy experiments. These methods for tracking G protein localization were also used to probe other questions. The activation of GPCRs from different classes had virtually no impact on the localization of G-beta, suggesting that GPCR activation does not result in the shuttling of G proteins through the endosomal pathway with activated receptors.Strengths:The question probed in this study is quite important and, in my opinion, understudied by the pharmacology community. The results presented here are an important call to be cognizant of the localization of GPCR coupling partners in different cellular compartments. Abundant reports of endosomal GPCR signaling need to consider how the impact of lower G protein abundance on endosomal membranes will affect the signaling responses under study.The work presented is carefully executed, with seemingly high levels of technical rigor. These studies benefit from probing the experimental questions at hand using two different methods of measurement (fluorescent microscopy and bystander BRET). The observation that both methods arrive at the same (or a very similar) answer inspires confidence about the validity of these findings.Weaknesses:The rationale for fusing G-beta with either mNG2(11) or SmBit could benefit from some expansion. I understand the speculation that using the smallest tag possible may have the smallest impact on protein performance and localization, but plenty of researchers have fused proteins with whole fluorescent proteins to provide conclusions that have been confirmed by other methods. Many studies even use G proteins fused with fluorescent proteins or luciferases. Is there an important advantage to tagging G-beta with small tags? Is there evidence that G proteins with full-size protein tags behave aberrantly? If the studies presented here would not have been possible without these CRISPR-based tagging approaches, it would be helpful to provide more context to make this clearer. Perhaps one factor would be interference from newly synthesized G proteins-fluorescent protein fusions en route to the plasma membrane (in the ER and Golgi).

There are several advantages to using small peptide tags that we did not fully explain. From a practical standpoint the most important advantage of using the HiBit tag instead of full-length Nanoluc is that it allows us to restrict luminescence output to cells transiently transfected with LgBit. In this way untransfected cells contribute no background signal. Although we did not take advantage of it here, this also applies to fluorescent protein complementation, and will be useful for visualizing proteins in individual cells within tissues. The HiBit tag also allows PAGE analysis by probing membranes with LgBit (as in Fig. 1). We are not aware of evidence that tagging Gb or Gg subunits on the N terminus results in aberrant behavior, while there is some evidence that Ga subunits tagged with full-size protein tags (in some positions) have altered functional properties (PMID: 16371464). We do think that editing endogenous genes is critical, as studies using transient overexpression (usually driven by strong promoters) have sometimes reported accumulation of tagged G proteins in the biosynthetic pathway (e.g., PMID: 17576765), as the reviewer suggests. Ga and Gbg appear to be mutually dependent on each other for appropriate trafficking to the plasma membrane (reviewed in PMID: 23161140), therefore the native (presumably matched) stoichiometry is likely to be critical.

To clarify this context the revised manuscript includes the following:

“For bioluminescence experiments we added the HiBit tag (Schwinn et al., 2018) and isolated clonal “HiBit-b1“ cell lines. An advantage of this approach over adding a full-length Nanoluc luciferase is that it requires coexpression of LgBit to produce a complemented luciferase. This limits luminescence to cotransfected cells and thus eliminates background from untransfected cells.”

“Some studies using overexpressed G protein subunits have suggested that a large pool of G proteins is located on intracellular membranes, including the Golgi apparatus (Chisari et al., 2007; Saini et al., 2007; Tsutsumi et al., 2009), whereas others have indicated a distribution that is dominated by the plasma membrane (Crouthamel et al., 2008; Evanko, Thiyagarajan, & Wedegaertner, 2000; Marrari et al., 2007; Takida & Wedegaertner, 2003). A likely factor contributing to these discrepant results is the stoichiometry of overexpressed subunits, as neither Ga nor Gbg traffic appropriately to the plasma membrane as free subunits (Wedegaertner, 2012). Our gene-editing approach presumably maintains the native subunit stoichiometry, providing a more accurate representation of native G protein distribution.”

As noted by the authors, they do not demonstrate that the tagged G-beta is predominantly found within heterotrimeric G protein complexes. If there is substantial free G-beta, then many of the conclusions need to be reconsidered. Perhaps a comparison of immunoprecipitated tagged G beta vs immunoprecipitated supernatant, with blotting for other G protein subunits would be informative.

We do think that HiBit-b1 exists predominantly within heterotrimeric complexes, for several reasons. First, overexpression studies have shown that Gbg requires association with Ga to traffic to the plasma membrane, and that by itself Gbg is retained on the endoplasmic reticulum

(PMID: 12609996; PMID: 12221133). We find almost no endogenous Gb1 on the endoplasmic reticulum, and a high density on the plasma membrane. Second, we are able to detect large increases in free HiBit-Gbg after G protein activation using free Gbg sensors (e.g. Fig. 1). Third, many proteins that bind to free Gbg are found entirely in the cytosol of HEK 293 cells (e.g. PMID: 10066824), suggesting there is not a large population of free Gbg. We have added discussion of these points to the revised manuscript as follows:

“Endogenous Ga and Gb subunits are expressed at approximately a 1:1 ratio, and Gb subunits are tightly associated with Gg and inactive Ga subunits (Cho et al., 2022; Gilman, 1987; Krumins & Gilman, 2006). Moreover, proteins that bind to free Gbg dimers are found in the cytosol of unstimulated HEK 293 cells, suggesting at most only a small population of free Gbg in these cells. Therefore, we assume that the large majority of mNG-b1 and HiBit-b1 subunits in unstimulated cells are part of heterotrimers.”

“Notably, when Gbg dimers are expressed alone they accumulate on the endoplasmic reticulum

(Michaelson et al., 2002; Takida & Wedegaertner, 2003). That we detect almost no endogenous Gbg on the endoplasmic reticulum supports our conclusion that the large majority of Gbg in unstimulated HEK 293 cells is associated with Ga, although we cannot rule out a small population of free Gbg.”

We do not entirely understand the suggested experiment, as free Gbg will still be largely associated with the membrane fraction. Notably, we find almost no HiBit-b1 in the supernatant after lysis in hypotonic buffer and preparation of membrane fractions, and the small amount that we do find does not change if Ga is overexpressed.

Additional context and questions:(1) There exists some evidence that certain GPCRs can form enduring complexes with G-betagamma (PubMed: 23297229, 27499021). That would seem to offer a mechanism that would enable receptor-mediated transport of G protein subunits. It would be helpful for the authors to place the findings of this manuscript in the context of these previous findings since they seem somewhat contradictory.

We agree. In our original submission we noted “It is possible that other receptors will influence G protein distribution using mechanisms not shared by the receptors we studied.” In the revised manuscript we have added:

“For example, a few receptors are thought to form relatively stable complexes with Gbg, which could provide a mechanism of trafficking to endosomes (Thomsen et al., 2016; Wehbi et al., 2013).”

(2) There is some evidence that GaS undergoes measurable dissociation from the plasma membrane upon activation (see the mechanism of the assay in PubMed: 35302493). It seems possible that G-alpha (and in particular GaS) might behave differently than the G-beta subunit studied here. This is not entirely clear from the discussion as it now stands.

Indeed, there is abundant evidence that some Gas translocates away from the plasma membrane upon activation. We referred to translocation of “some Ga subunits” in the introduction, although we did not specify that Gas is by far the most studied example. In a previous study (PMID: 27528603) we found that overexpressed Gas samples many intracellular membranes upon activation and returns to the plasma membrane when activation ceases. This is similar to activation-dependent translocation of free Gbg dimers. Because these translocation mechanisms depend on activation and are reversible they are unlikely to be a major source of inactive heterotrimers for intracellular membranes.

We did a poor job of making it clear that we intentionally avoided translocation mechanisms that operate only *during* receptor and G protein stimulation. In the revised manuscript we have added new data showing reversible activation-dependent translocation of endogenous HiBitGb1.

(3) The authors say "The presence of mNG-b1 on late endosomes suggested that some G proteins may be degraded by lysosomes". The mechanism of lysosomal degradation by proteins on the outside of the lysosome is not clear. It would be helpful for the authors to clarify.

We agree we didn’t connect the dots here. Our initial idea was that G proteins on the surface of late endosomes might reach the interior of late endosomes and then lysosomes by involution into multivesicular bodies. However, the reviewer correctly points out that much of the G protein associated with lysosomes still appears to be on the cytosolic surface, where it would not be subject to degradation. In fact, since lysosomes can fuse with the plasma membrane under certain circumstances, this could even represent a pathway for recycling G proteins to the plasma membrane.

We have revised the text to avoid giving the impression that lysosomes degrade G proteins, since we have scant evidence that this occurs. In the revised discussion we point out that we do not know the fate of G proteins located on the surface of lysosomes and speculate that these could be returned to the plasma membrane:

“We do not know the fate of G proteins located on the surface of lysosomes. Since lysosomes may fuse with the plasma membrane under certain circumstances (Xu & Ren, 2015), it is possible that this represents a route of G protein recycling to the plasma membrane.”

(4) Although the authors do a good job of assessing G protein dilution in endosomal membranes, it is unclear how this behavior compares to the measurement of other lipidanchored proteins using the same approach. Is the dilution of G proteins what we would expect for any lipid-anchored protein at the inner leaflet of the plasma membrane?

This is a great question. To begin to address it we have studied a model lipid-anchored protein consisting of mNeongreen2 anchored to the plasma membrane by the C terminus of HRas, which is palmitoylated and prenylated. We find that this protein is also diluted on endocytic vesicles, although to a lesser degree than heterotrimeric G proteins. We have added a section to the results and a new figure supplement describing these results:

“To test if other peripheral membrane proteins are similarly depleted from endocytic vesicles, we performed analogous experiments by overexpressing mNG bearing the C-terminal membrane anchor of HRas (mNG-HRas ct). We found that mNG-HRas ct was also less abundant on FM464-positive endocytic vesicles than expected based on plasma membrane abundance, although not to the same extent as mNG-b1 (Figure 4 - figure supplement 2); mNG-HRas ct density on FM4-64-positive vesicles was 64 ± 17% (mean ± 95% CI; n=78) of the nearby plasma membrane.”

**Reviewer #2 (Public Review):**
This is an interesting method that addresses the important problem of assessing G protein localization at endogenous levels. The data are generally convincing.Specific commentsMethods:The description of the gene editing method is unclear. There are two different CRISPR cell lines made in two different cell backgrounds. The methods should clearly state which CRISPR guides were used on which cell line. It is also not clear why HiBit is included in the mNG-β1 construct. Presumably, this is not critical but it would be helpful to explicitly note. In general, the Methods could be more complete.

We have added the following to the methods to clarify that the same gRNA was used to produce both cell lines:

“The human GNB1 gene was targeted at a site corresponding to the N-terminus of the Gb1 protein; the sequence 5’-TGAGTGAGCTTGACCAGTTA-3’ was incorporated into the crRNA, and the same gRNA was used to produce both HiBit-b1 and mNG-b1 cell lines.”

We have added the following to the methods to clarify why HiBit is included in the mNG-b1 construct:

“HiBit was included in the repair template for producing mNG-b1 cells to enable screening for edited clones using luminescence.”

Results:The explanation of validation experiments in Figures 1 C and D is incomplete and difficult to follow. The rationale and explanation of the experiments could be expanded. In addition, because this is an interesting method, it would be helpful to know if the endogenous editing affects normal GPCR signaling. For example, the authors could include data showing an Isoinduced cAMP response. This is not critical to the present interpretation but is relevant as a general point regarding the method. Also, it may be relevant to the interpretation of receptor effects on G protein localization.

We have expanded the rationale and explanation of experiments in Figures 1C and D by adding:

“For example, we observed agonist-induced BRET between the D2 dopamine receptor and mNG-b1, an interaction that requires association with endogenous Ga subunits (Figure 1C). Similarly, we observed BRET between HiBit-b1 and the free Gbg sensor memGRKct-Venus after activation of receptors that couple Gi/o, Gs, and Gq heterotrimers, indicating that HiBit-b1 associated with endogenous Ga subunits from these three families (Figure 1D).”

We have done the suggested cAMP experiment and provide the data in a new figure supplement:

“We also found that cyclic AMP accumulation in response to stimulation of endogenous b adrenergic receptors was similar in edited cell lines and their unedited parent lines (Figure 1 - figure supplement 1).”

Discussion:The conclusion that beta-gamma subunits do not redistribute after GPCR activation seems new and different from previous reports. Is this correct? Can the authors elaborate on how the results compare to previous literature?

Many previous studies have indeed shown that *free* Gbg dimers can redistribute after GPCR activation and sample intracellular membranes. Our initial focus was on possible changes in heterotrimer distribution after GPCR activation, but in retrospect we should have directly addressed free Gbg translocation and made the distinction clear.

In the revised manuscript we show that *during* stimulation we observe changes consistent with modest translocation of endogenous Gbg from the plasma membrane and sampling of intracellular compartments. To our knowledge this is the first demonstration of endogenous Gbg translocation.

We have added:

“With overexpressed G proteins free Gbg dimers translocate from the plasma membrane and sample intracellular membrane compartments after activation-induced dissociation from Ga subunits. Consistent with this, we observed small decreases in bystander BRET at the plasma membrane and small increases in bystander BRET at intracellular compartments during activation of GPCRs, suggesting that endogenous Gbg subunits undergo similar translocation (Figure 5- figure supplement 1). Notably, these changes occurred at room temperature, suggesting that endocytosis was not involved, and developed over the course of minutes. The latter observation and the small magnitude of agonist-induced changes are both consistent with expression of primarily slowly-translocating endogenous Gg subtypes in HEK 293 cells. Moreover, as shown previously for overexpressed Gbg, the changes we observed with endogenous Gbg were readily reversible (Figure 5- figure supplement 1), suggesting that most heterotrimers reassemble at the plasma membrane after activation ceases.”

Can the authors note that OpenCell has endogenously tagged Gβ1 and reports more obvious internal localization? Can the authors comment on this point?

OpenCell has tagged GNB1 and the Leonetti group kindly provided a parent cell line we used to add a slightly different tag. Although their study did not identify any specific intracellular compartments, our impression is that most of the internal structures visible in their images are likely to be lysosomes, as they are large, round and often have a clear lumen. Overall their images and ours are comfortingly similar. We have added:

“Unsurprisingly, our images are quite similar to those made as part of previous study that labeled Gb1 subunits with mNG2 (Cho et al., 2022).”

Notably, the Leonetti group has recently reported the subcellular distribution of many untagged proteins using a proteomic approach. They find that Gb1 is enriched on the plasma membrane and lysosomes but is not enriched on endosomes, the Golgi apparatus, endoplasmic reticulum or mitochondria (https://www.biorxiv.org/content/10.1101/2023.12.18.572249v1). We have cited this work in the revised manuscript.

Is this the first use of CRISPR / HiBit for BRET assay? It would be helpful to know this or cite previous work if not. Also, as this is submitted as a tools piece, the authors might say a little more about the potential application to other questions.

The only previous study we are aware of utilizing a similar combination of methods is a 2020 report from the group of Dr. Stephen Hill, in which the authors studied binding of fluorescent ligands to HiBit-tagged GPCRs. This work is now cited.

We have also added the following to our previous brief statement about potential applications:

“In addition, it may also be possible to use these cells in combination with targeted sensors to study endogenous G protein activation in different subcellular compartments. More broadly, our results show that subcellular localization of endogenous membrane proteins can be studied in living cells by adding a HiBit tag and performing bystander BRET mapping. Applied at large scale this approach would have some advantages over fluorescent protein complementation, most notably the ability to localize endogenous membrane proteins that are expressed at levels that are too low to permit fluorescence microscopy.”

**Reviewer #3 (Public Review):**
Summary:This article addresses an important and interesting question concerning intracellular localization and dynamics of endogenous G proteins. The fate and trafficking of G protein-coupled receptors (GPCRs) have been extensively studied but so far little is known about the trafficking routes of their partner G proteins that are known to dissociate from their respective receptors upon activation of the signaling pathway. The authors utilize modern cell biology tools including genome editing and bystander bioluminescence resonance energy transfer (BRET) to probe intracellular localization of G proteins in various membrane compartments in steady state and also upon receptor activation. Data presented in this manuscript shows that while G proteins are mostly present on the plasma membrane, they can be also detected in endosomal compartments, especially in late endosomes and lysosomes. This distribution, according to data presented in this study, seems not to be affected by receptor activation. These findings will have implications in further studies addressing GPCR signaling mechanisms from intracellular compartments.Strengths:The methods used in this study are adequate for the question asked. Especially, the use of genome-edited cells (for the addition of the tag on one of the G proteins) is a great choice to prevent the effects of overexpression. Moreover, the use of bystander BRET allowed authors to probe the intracellular localization of G proteins in a very high-throughput fashion. By combining imaging and BRET authors convincingly show that G proteins are very low abundant on early endosomes (also ER, mitochondria, and medial Golgi), however seem to accumulate on membranes of late endosomal compartments.Weaknesses:While the authors provide a novel dataset, many questions regarding G protein trafficking remain open. For example, it is not entirely clear which pathway is utilized to traffic G proteins from the plasma membrane to intracellular compartments. Additionally, future studies should also address the dynamics of G protein trafficking, for example by tracking them over multiple time points.

We agree, there is much more to do.

**Recommendations for the authors:**

**Reviewer #1 (Recommendations For The Authors):**
On page 7 the text says "the difference did reach significance (Figure 5D)". It looks like the difference did not reach significance. Please check on this.

Thank you, this was an unfortunately significant typo.

**Reviewer #3 (Recommendations For The Authors):**
This article addresses an important and interesting question concerning intracellular localization and dynamics of endogenous G proteins. While the posed question is indeed a grand one and the methods used by the authors are novel, I believe that the data presented in this manuscript are still insufficient to support all claims posed by the authors. Below I list my major concerns:(1) The authors claim that they provide a "detailed subcellular map of endogenous G protein distribution", however, the map is in my opinion not sufficiently detailed (e.g. trans-Golgi network is not included) and not quantitative enough (e.g. % of proteins present on one compartment vs. the other as authors claim that BRET signals "cannot be directly compared between different compartments"). To strengthen this statement, except for providing more extensive and quantitative data, it would be beneficial to provide such a "map" as an illustration based on the findings presented in this article.

“Detailed” is certainly a subjective term. While we maintain that our description of endogenous G protein distribution is far more detailed than any previous study, we now simply claim to provide a “subcellular map”. We have added images of TGNP (TGN46; TGOLN2), showing that endogenous G proteins are readily detectable on the structures labeled by this marker. These data are now provided in Figure 3 – figure supplement 7.

We did not claim that our study was quantitative- we did not try to count G proteins. However, if we use published estimates of total G proteins and surface area for HEK 293 cells we estimate that there are roughly 2,500 G proteins µm-2 on the plasma membrane and 500 G proteins µm-2 on endocytic vesicles. For other intracellular compartments relative density can be approximated by inspecting images, but a truly quantitative estimate would require a surface area standard analogous to FM4-64 for each compartment. The percentage of the total G protein pool on a given compartment is, in our opinion, less important than the density of G proteins on that compartment, as the latter is more likely to affect the efficiency of local signal transduction. Since we do not claim to have accurate G protein density estimates for many intracellular compartments, we prefer to provide several raw images for each compartment rather than a schematized map.

Bystander BRET values cannot be compared directly across compartments due to differences in expression and energy transfer efficiency of different markers and compartment surface area. This method is well suited for following changes in distribution as a function of time or after perturbations and for sensitive detection of weak colocalization but can only provide approximate “maps” of absolute distribution.

(2) Probing of the intracellular distribution of these proteins, especially after GPCR activation, includes a single chosen timepoint. I believe that the manuscript would greatly benefit from including some dynamic data on internalization and intracellular trafficking kinetics. What is the turnover of tested G proteins? What is the fraction that is going to recycling compartments and/or lysosomes? Authors could perhaps turn to other methods to be able to dynamically track proteins over time e.g. via photoconversion techniques.

Because G protein trafficking appears to be largely constitutive there is no easy way for us to assess how long it takes G proteins to transit various intracellular compartments, although we agree this would be interesting. As the reviewer suggests, dynamic data on constitutive trafficking would require methods (such as photoconversion) not currently available to us for endogenous G proteins. Accordingly, we have made no claims regarding the kinetics of G protein trafficking. As for possible redistribution after GPCR activation, in the revised manuscript we have added 5- and 15-minute timepoints after agonist stimulation for our bystander BRET mapping (Figure 5- figure supplement 2). These timepoints were chosen to correspond to persistent signaling mediated by internalized receptors.

(3) Exemplary images with cells showing significant colocalization with lysosomal compartments seem to contain more intracellular vesicles visible in the mNG channel than in the case of the other compartment. Is it an effect of the treatment to stain lysosomes? It would be beneficial to compare it with some endogenous marker e.g. LAMP1 without additional treatments.

The visibility of intracellular vesicles in our lysosome images likely reflects our selection of cells and regions with visible and abundant lysosomes, specifically peripheral regions directly adhered to the coverslip, rather than treatment with lysosomal stains (LV 633 and dextran). As suggested, we now include images of cells expressing LAMP1 as an alternative lysosome marker (Figure 3 - figure supplement 6).

(4) The authors probe an abundance of G proteins along the constitutive endocytic pathway. However, to prove that G proteins are not de-palmitoylated rather than endocytosed authors should perform control experiments where endocytosis is blocked e.g. pharmacologically or via a knockdown approach. Additionally, various endocytic pathways can be probed.

We did not claim that depalmitoylation plays no role in delivery of G proteins to internal compartments. In fact, we pointed out that we cannot at present rule out other pathways and delivery mechanisms. Importantly, if some of the G proteins that we detect along the endocytic pathway do arrive there by trafficking through the cytosol this would only strengthen our major conclusion that endocytosis is inefficient.

Having said this, we have now conducted extensive experiments investigating the role of palmitate cycling in the trafficking of heterotrimeric G proteins and the small G protein H-Ras. Our results suggest that a depalmitoylation-repalmitoylation cycle is not important for the distribution of heterotrimers, but these findings will be the subject of a separate publication focused on this specific question for both large and small G proteins.

We agree that it will be interesting to probe different endocytic pathways, as suggested using a genetic approach. Our main interest here was in endocytic membranes that were defined functionally (with FM4-64 or internalized receptors) rather than biochemically.

Minor comments:(5) "Imaging" paragraph in the Methods section refers to a non-existent figure called "SI Appendix S9".

Thank you.

(6) It is not clear what was used as a "control" in Figure 5E.

“Control” refers to DPBS vehicle alone. This information is now added to the legend for Figure 5E.